# The Role of Functionalization and Size of Gold Nanoparticles in the Response of MCF-7 Breast Cancer Cells to Ionizing Radiation Comparing 2D and 3D In Vitro Models

**DOI:** 10.3390/pharmaceutics15030862

**Published:** 2023-03-07

**Authors:** Marika Musielak, Agnieszka Boś-Liedke, Oliwia Piwocka, Katarzyna Kowalska, Roksana Markiewicz, Barbara Szymkowiak, Paweł Bakun, Wiktoria M. Suchorska

**Affiliations:** 1Department of Electroradiology, Poznan University of Medical Sciences, 61-866 Poznan, Poland; 2Doctoral School, Poznan University of Medical Sciences, 60-812 Poznan, Poland; 3Radiobiology Laboratory, Department of Medical Physics, Greater Poland Cancer Centre, 61-866 Poznan, Poland; 4Department of Macromolecular Physics, Faculty of Physics, Adam Mickiewicz University, 61-614 Poznan, Poland; 5Department of Histology and Embryology, Poznan University of Medical Sciences, 61-781 Poznan, Poland; 6NanoBioMedical Centre, Adam Mickiewicz University, 61-614 Poznan, Poland; 7Faculty of Physics, Adam Mickiewicz University, 61-614 Poznan, Poland; 8Chair and Department of Chemical Technology of Drugs, Poznan University of Medical Sciences, 61-781 Poznan, Poland

**Keywords:** Au nanoparticles, metal nanoparticles, radiotherapy, radiosensitization, oncology

## Abstract

Gold nanoparticles (AuNPs), as an agent enhancing radiosensitivity, play a key role in the potential treatment of breast cancer (BC). Assessing and understanding the kinetics of modern drug delivery systems is a crucial element that allows the implementation of AuNPs in clinical treatment. The main objective of the study was to assess the role of the properties of gold nanoparticles in the response of BC cells to ionizing radiation by comparing 2D and 3D models. In this research, four kinds of AuNPs, different in size and PEG length, were used to sensitize cells to ionizing radiation. The in vitro viability, uptake, and reactive oxygen species generation in cells were investigated in a time- and concentration-dependent manner using 2D and 3D models. Next, after the previous incubation with AuNPs, cells were irradiated with 2 Gy. The assessment of the radiation effect in combination with AuNPs was analyzed using the clonogenic assay and γH2AX level. The study highlights the role of the PEG chain in the efficiency of AuNPs in the process of sensitizing cells to ionizing radiation. The results obtained imply that AuNPs are a promising solution for combined treatment with radiotherapy.

## 1. Introduction

Nanotechnology is a field of science that studies structures who have at least one dimension less than or equal to 100 nm [1]. The most important feature of materials in the nanometric scale is the transition in physicochemical properties along with the change in their size [2]. Gold nanoparticles (Au nanoparticles, AuNPs) are often used to sensitize cancer cells to ionizing radiation [3]. These are characterized by a high surface to volume ratio, the possibility of surface modification, and specific optical properties [4]. The surface modification of AuNPs is highly related to the aim to be achieved (e.g., AuNPs have been chemically modified with low molecular branched polyethylenimine for the efficient delivery of gapmers targeting p53 mutant protein) [5]. Xue et al. designed a multi-functional three-dimensional (3D) DNA shell consisting of DNA bricks with sticky ends (sticky-YTDBs) and tiled them onto siRNA-packaged AuNPs [6]. Recent advances in the multi-functional design of AuNPs enable the generation of localized heat in the cancer tissues as well as the controlled and targeted delivery of multiple desired drugs. AuNPs have numerous advantages that make them suitable for photothermal or irradiation treatment [7]. Introducing AuNPs into the tumor area is a potential strategy to solve the problem of protecting healthy tissues in the beam field. Sensitizing tumor cells to radiotherapy would allow for lower radiation doses, reducing the negative effects of radiation on healthy tissues. This involves still unknown physical, chemical, and biological processes that require further research. Butterworth et al. [8] performed a time course experiment to determine the effect of AuNPs on cell survival in the absence of radiation and the optimal nanoparticle incubation period prior to irradiation. The implementation of 1.9 nm gold particles caused a variety of cell line-specific responses such as decreased clonogenic survival, increased apoptosis, and DNA damage induction, which may be mediated by the production of reactive oxygen species. It was the first study using 1.9 nm sized particles to report multiple cellular responses that influence the radiation dose modifying effect. This article emphasized the importance of thoroughly characterizing responses to gold nanoparticles when assessing dose-enhancing potential in cancer therapy. The radiosensitization effect of glucose-capped AuNPs (Glu-AuNPs) of different sizes (16 nm and 49 nm) on MDA-MB-231 cells in the presence of megavoltage X-rays was also described [9]. The authors discovered that Glu-AuNPs could increase the radiosensitivity toward BC cells, most likely by regulating the cell cycle distribution, with a higher number of cells in the G2/M phase. They highlighted that the amount of Glu-AuNPs in the cells might be related to the effect of radiation enhancement. Due to the wide range of surface functionalization possibilities of AuNPs, there is a noticeable lack of systematic analysis in the literature, which does not allow for a proper comparison of the experimental results. There is a need for research groups to focus on improving the already proposed strategies and attempts to conduct experiments in multidimensionally controlled experimental conditions.

Another important aspect is the method of evaluating the effectiveness of nanoparticles depending on the research model used. Promising results obtained in 2D conditions are often not reflected in the results of in vivo studies using animal models [10]. 2D cultures are devoid of a three-dimensional network of different cell types surrounded by an extracellular matrix. These two aspects play a crucial role in the diffusion and uptake of nanoparticles by cells. However, the evaluation of their performance remains limited as most work is carried out on two-dimensional culture systems.

The main objective of this work was to assess the role of gold nanoparticle properties in the response of breast cancer cells to ionizing radiation in 2D and 3D models. Specific objectives included (1) the synthesis, functionalization, and characterization of spherical gold nanoparticles; (2) the evaluation and comparison of internalization of nanoparticles with different properties between cells in 2D and 3D culture; and (3) the evaluation of the impact of the incubation time of gold nanoparticles with breast cancer cells in 2D and 3D culture on the response of cells after exposure to ionizing radiation.

## 2. Materials and Methods

### 2.1. Methods

#### 2.1.1. Synthesis of AuNPs

Nanospheres were prepared using two techniques: the inverse method published by Schulz et al. [11] and the Turkevich technique presented by Wang et al. [12].

The preparation procedure of 10 nm nanospheres:

According to Schulz et al. [11], presenting the inverse method, 25.8 mg NaCl (trisodium citrate dihydrate) was dissolved in 50 mL of water. Then, 1 mL of EDTA (3.8 mg/mL, ethylenediaminetetraacetic acid) was added to the solution and boiled for 15 min before precursor addition. The precursor solution (8.49 mg/mL HAuCl_4_·3H_2_O, gold(III) chloride) was prepared in a separate flask according to the calculated molar ratio (MR). After boiling, 1 mL of precursor solution was injected while the mixture was rapidly stirred. When the color changed to wine-red, which indicated the formation of AuNPs, the heating was switched off. The solution was removed when the temperature reached 70 °C. The solution was filtered. Using this technique, only 10 ± 2 nm nanospheres could be synthesized with MR = 4. The initial concentration of AuNPs_10nm_ was 0.277 mg/mL. 

The preparation procedure of 30 nm nanospheres:

The 30 nm nanospheres were synthesized according to Wang et al. [12] where 25 mL of water was stirred in a flask and heated to 100 °C. Separately, 52.4 mg of HAuCl_4_ was dissolved in 75 mL of water, and 10 mL of the solution was poured into the hot water flask. Afterward, the citrate stock was prepared by dissolving 506.3 mg of NaCt in 75 mL of water. Then, 4 mL of the stock solution was added to HAuCl_4_ and boiled for 1.5 h. Next, the solution was left on the table until the temperature reached room temperature. The initial concentration of AuNPs_30nm_ was 0.286 mg/mL. 

#### 2.1.2. Functionalization of AuNPs

For the functionalization of 10 nm and 30 nm nanospheres, the RGD (Ary–Gly–Asp) complex and polyethylene glycol (PEG) chains were used in two molecular weights of 800 and 2000. The functionalization was performed according to the protocol by Yang et al. [13]. We obtained four nanostructures: RGD-PEG_800_-AuNPs_10nm_, RGD-PEG_2000_-AuNPs_10nm_, RGD-PEG_800_-AuNPs_30nm_, and RGD-PEG_2000_-AuNPs_30nm_. 

#### 2.1.3. Characterization of AuNPs

The morphology of the synthesized AuNPs with sizes of 10 nm and 30 nm was analyzed by transmission electron microscopy (TEM). On a copper grid covered in a Formvar-carbon membrane (300 mesh, Ted Pella Inc., Redding, CA, USA)), 5–10 L of an aqueous dispersion was first applied and air dried at room temperature. Ultrastructural and morphometric analysis of the gold nanoparticles was performed using a TEM model JEM 1010 (Jeol, Tokyo, Japan) equipped with a MEGAVIEW G2 camera and the cooperating iTEM Digital Imaging Solutions program (Olympus Soft Imaging Solutions GmbH, Münster, Germany). The average diameter of the AuNPs reached about 10.5 nm +/− 1.3 nm and 30.9 nm +/− 1.8. The UV–VIS measurements were performed using a Jasco 650v (ABL&E-JASCO, Kraków, Poland).

### 2.2. Cell Culture

#### 2.2.1. 2D Model

The breast cancer cell line MCF-7 was used in the study (ATCC). The standard conditions for cell maintenance were a 37 °C incubator (Binder, Tuttlingen, Germany) with a water vapor-saturated atmosphere that was 5% CO_2_ enriched. Penicillin/streptomycin (P/S) at a final concentration of 1% (Merck Millipore Corporation, Darmstadt, Germany) and 0.01 mg/mL of insulin (Gensulin R, Bioton, Macierzysz, Poland) were added to DMEM (Biowest, Nuaillé, France) supplemented with 10% fetal bovine serum (FBS) (Biowest, France) for the MCF-7 culture medium. Cells were passaged with trypsin-EDTA when confluency reached 80–90%. Tests were executed using a laminar flow hood under aseptic conditions.

#### 2.2.2. 3D Model

MCF-7 cells were seeded on the U-bottom 96-well in the concertation of 6000 cells per well. After 6 days, the mammospheres were ready for the experiments. The cell culture medium and cell culture conditions were the same as for the 2D model.

### 2.3. Live/Dead Assay

Cells were seeded at concentrations of 4000, 6000, and 8000 per well. The staining solution consisted of 2.5 µL calcein AM, 10 µL ethidium homodimer-1, and 5 mL of PBS. The medium was removed from the cells, and wells were washed with PBS. Then, 100 µL of the staining solution was added directly to cells and incubated for 30 min at room temperature. Afterward, the cells were imaged on an Olympus IX83 inverted fluorescence microscope (Olympus, Tokyo, Japan).

### 2.4. Viability Test

Cells were plated at a concentration of 15,000 cells per well (MDA-MB-231 and MCF12A) onto 96-well flat-bottomed plates. After 24 h, RGD-PEG_800_-AuNPs_10nm_, RGD-PEG_2000_-AuNPs_10nm_, RGD-PEG_800_-AuNPs_30nm_, and RGD-PEG_2000_-AuNPs_30nm_ were added at 0.0004, 0.0008, 0.0012, 0.0020, 0.0060, 0.0120, and 0.0200 mg/mL concentrations and distributed in culture media at final volumes of 100 μL per well. Tests were performed for 24 and 48 h of cell incubation. The culture media containing AuNPs were then removed, and MTT (3-(4,5-dimethylthiazol-2-yl)-2,5-diphenyltetrazolium bromide) (Affymetrix, Cleveland, OH, USA) was added at a final concentration of 0.5 mg/mL. The medium was removed after 3 h of cell culture, and 100 μL of DMSO (Thermo Scientific, Waltham, MA, USA) was then added to each well to dissolve the formazan crystals that had formed. A Multiskan plate reader (Thermo Scientific, Waltham, MA, USA) at 570/590 nm, background 655 nm, was used to read the outcome. 

### 2.5. Reactive Oxygen Species (ROS) Assay

A total of 200,000 cells per well were used for cell seeding. In the culture medium devoid of serum, a 1:1000 solution of 2′,7′-dichlorofluorescin diacetate DCFH-DA (Merck Millipore Corporation, Germany) was made. Cells were PBS-washed once after the culture media were removed. After that, a dye solution was applied to the cells, and then incubated at 37 °C for 45 min. Following this, the dye solution was removed, and cells underwent one PBS wash. Complete culture medium mixed with each AuNP—RGD-PEG_800_-AuNPs_10nm_, RGD-PEG_2000_-AuNPs_10nm_, RGD-PEG_800_-AuNPs_30nm_ and RGD-PEG_2000_-AuNPs_30nm_—at a concentration of 0.0004, 0.0008, and 0.0012 mg/mL was added to the cells followed by incubation of 10, 20, 30, 40 min and 1, 3, 6, and 24 h. Cells were gathered, suspended in PBS, and subjected to flow cytometric analysis. Fluorescein isothiocyanate-height (FITC-H) channel measurements were applied using a flow cytometer (at 485 nm excitation wavelength and 527 nm emission wavelength). 

### 2.6. Internalization Analysis

The uptake of the AuNPs was measured according to the method published by Park et al. [14]. Cells were seeded on the 12-well plates (200,000 cells/well). After 24 h, RGD-PEG_800_-AuNPs_10nm_, RGD-PEG_2000_-AuNPs_10nm_, RGD-PEG_800_-AuNPs_30nm_, and RGD-PEG_2000_-AuNPs_30nm_ were added in the concentration of 0.0004, 0.0008, and 0.0012 mg/mL for each type of AuNP. The internalization level of AuNPs in the cells was investigated after 0.5, 1, 3, 6, and 24 h. After incubation, cells were harvested, suspended in phosphate buffered saline (PBS) (Biowest, France) and washed once to discard excess AuNPs from the sample. A cytometric analysis was conducted using a Cytoflex Beckmann Coulter cytometer (Beckman Coulter Life Sciences, Indianapolis, IN, USA). By examining fluorescence at 611 nm, the side scatter parameter (SSC) was determined. FlowJo v10 was used for the analysis of the outcomes. 

### 2.7. Irradiation

Cells were irradiated after incubation with RGD-PEG_800_-AuNPs_10nm_, RGD-PEG_2000_-AuNPs_10nm_, RGD-PEG_800_-AuNPs_30nm_, and RGD-PEG_2000_-AuNPs_30nm_. Three different time points (TPs) of cells incubated with AuNPs were chosen based on the previous results: 30 min, 3 h, and 24 h. First, the cells were plated on 10 cm plates. Then, a medium with one type of AuNP was added to the cells and incubated for the chosen TP. After that, the cells were gathered and exposed to radiation. Using a closed source of Cs-137 with an activity of 20.4 TBq, the radiation was administered using a Gamma Cell^®^ 1000 Elite device (BestTheratronics Ltd., Kanata, ON, Canada) at a dose rate of 2.5 Gy/min, and 2 Gy of radiation was applied to the cells. 

### 2.8. Clonogenic Assay

After irradiation, the optimized number of cells was plated on 6-well plates and incubated for 7 days. Clonogenic assays were closed when a colony that reached a minimum of 50 clones was observed in the control group. The medium was discarded, and cells were washed with PBS. Next, the fixation of cells was performed with denatured ethanol. Following the removal of the ethanol, the plates were stained roughly with 2 mL of Coomassie Blue solution (Merck Millipore Corporation, Germany) and left to develop for 20 min. The plates were then washed in warm water, dried, and the buffer was discarded. The ChemiDoc Touch Bio-Rad system (Hercules, Clearwater, FL, USA) was used to take pictures of the plates. The Gene Tools Syngene program was applied to complete an automatic colony counting. The clonogenic assay on the 3D model was conducted in the same conditions as for the 2 D model. The difference was that the incubation with AuNPs before irradiation was performed in the 3D model. The culture medium was replaced with the fresh medium and AuNPs in the selected concentration. Next, spheres were collected and partially resuspended for irradiation. Further procedures where the same in both models. The spheres were dissociated into single cells for plating. To assess the radiosensitivity effect of AuNPs after irradiation, the group incubated with AuNPs only was applied for the calculation as a control to the group treated with AuNPs + 2 Gy.

### 2.9. Flow Cytometry Analysis after Irradiation—γH2AX

After irradiation, cells were separated into samples of 200,000 cells. After 45 min, cells were washed with PBS, then fixed and permeabilized with the Fixation/Permeabilization Kit (BD Biosciences, Franklin Lakes, NJ, USA). Cells were incubated with the 3.5 µL anti- γH2AX antibody (Becton Dickinson, Franklin Lakes, NJ, USA) and PBS in the final volume of 20 µL for 30 min at 4 °C. Finally, cells were washed with PBS and suspended in 200 µL of PBS for the cytometric analysis. 

### 2.10. Immunofluorescence

The immunofluorescence was completed for γH2AX after irradiation of the cells. Cells were seeded on 8-well chamber slides (VWR, Darmstadt, Germany) with a 50,000 cells/well density. After 24 h, a solution of the culture medium with each AuNP—RGD-PEG_800_-AuNPs_10nm_, RGD-PEG_2000_-AuNPs_10nm_, RGD-PEG_800_-AuNPs_30nm_ and RGD-PEG_2000_-AuNPs_30nm_—at a concentration of 0.0008 mg/mL was added to the cells. The 50 min TP was chosen for the incubation time of the cells with AuNPs. Following each TP incubation with AuNPs, the cells were irradiated with a dose of 2 Gy. After irradiation, the cells were incubated for 45 min at 37 °C. Next, the cells were washed with PBS, fixed in 4% paraformaldehyde for 20 min at room temperature (RT), and permeabilized with ice-cold 100% methanol at −20 °C for 20 min. Next, the blocking was performed by incubation with 0.2% Triton X-100 and 1% bovine serum albumin (BSA) (VWR, Germany) solution for 30 min at RT. After blocking, cells were washed with PBS. Next, 200 µL of a primary antibody solution γH2AX (ab22551, Abcam, Cambridge, UK) produced in mice was added into each chamber and slides were incubated overnight at 4 °C. After incubation, the cells were washed thrice with 2% BSA in PBS solution and incubated with 250 µL of secondary antibody solution (Jackson Immuno Research Labs, West Grove, PA, USA) for 1 h at 37 °C in darkness. All slides were washed thrice with 2% BSA in PBS solution, and 400 µL of DAPI (catalogue number: SAFSD8417) (VWR, Germany) solution was added. Immunofluorescence was imaged using an Olympus IX83 microscope (Boston Industries, Inc., Walpole, MA, USA). The quantification of γH2AX was performed using image deconvolution. To be able to compare the results, the parameter of the mean fluorescence intensity of γH2AX foci was used in both cell models. In the 2D model, the results were calculated as a ratio of the mean foci intensity per nuclei whereas in the 3D model, it was the mean foci intensity per area [μm^2^]. 

### 2.11. Statistical Analysis

The statistical analysis was performed using PQStat Software v.1.8.2. and Microsoft^®^ Excel^®^ (Microsoft Office Professional Plus 2019). The normality of the observed data distribution was assessed using the Shapiro–Wilk test. The one-way ANOVA was conducted for multiple comparisons. If Levene’s test indicated that the variances are not equal across the groups, the unequal variance *t* Test (Welch’s *t* Test) was implemented. To calculate the differences for a complex system (more than two groups), multiple comparison procedures were used by applying Tuckey’s post hoc test. The data were deemed significant at *p* < 0.05. The setting of the *p*-value was * *p* < 0.05, ** *p* < 0.01, *** *p* < 0.001. 

## 3. Results

### 3.1. Synthesis, Functionalization, and Characterization of AuNPs

The first specific objective was to synthesize, functionalize, and characterize the spherical gold nanoparticles (Figure 1). Two sizes, 10 nm and 30 nm, of AuNPs were obtained. Nanospheres were prepared using two techniques: the inverse method published by Schulz et al. [11] for 10 nm and the Turkevich technique presented by Wang et al. [12] for 30 nm. The primary studies using the protocol by Schultz et al. focused on the most suitable NaCt (trisodium citrate dihydrate) to HAuCl_4_·3H2O molar ratio MR choice. For that reason, the NaCt concentration was modified to achieve MR from 0.5 to 15 and the following synthesis was repeated several times. Using this technique only, 10 ± 2 nm nanospheres were able to be synthesized with MR = 4. The diameter and the shape of the synthesized nanostructures were checked using UV–VIS and TEM. Using the protocol by Yang et al. [13] for the functionalization oof AuNPs, we obtained four different nanostructures: RGD-PEG_800_-AuNPs_10nm_, RGD-PEG_2000_-AuNPs_10nm_, RGD-PEG_800_-AuNPs_30nm_, and RGD-PEG_2000_-AuNPs_30nm_. 

### 3.2. Cell Culture—2D and 3D Models

For the experiments, the MCF-7 breast cancer cell line was used. This cell line is characterized by the Luminal A subtype expressing estrogen receptor (ER) and progesterone receptor (PR) [15]. The luminal-A is the most common subtype and represents 50–60% of all breast cancers [16]. For the 3D model optimization, the live/dead assay was used to compare three numbers of seeded cells: 4000, 6000, and 8000 cells and times of formation of 5 and 7 days. The middle number of cells was chosen, and 7 days was the most effective scheme considering different incubation times in further experiments and the maximum size of the sphere, enabling us to imitate the minimum physiology of breast cancer tumors. After 7 days of forming, the breast cancer spheres were ready to use in experiments (Figure 2). 

### 3.3. Viability Assay

The MTT assay was evaluated for assessing the MCF-7 cell viability after incubation with AuNPs. In this test, RGD-PEG_800_-AuNPs_10nm_, RGD-PEG_2000_-AuNPs_10nm_, RGD-PEG_800_-AuNPs_30nm_, and RGD-PEG_2000_-AuNPs_30nm_ were used in a concentration of 0.0004, 0.0008, 0.0012, 0.0020, 0.0060, 0.0120, and 0.0200 mg/mL. To observe a wide range of effects, the TPs of 24, 48, and 72 h of the cell incubation time with AuNPs were chosen. All experiments were performed in triplicate in both models; 2D and 3D cell cultures. In the 2D model, the highest heterogeneity in results was observed in the smallest concentrations of AuNPs (Figure 3). The stability trend was noticed with a higher concentration of AuNPs. The lowest viability level was about 65% after 48 h of incubation with AuNPs of 10 nm in size. Statistically significant differences were not observed when comparing the viability results of all four nanostructures. The viability maintained a stable trend, reaching about 80–90% in all incubation times. The MTT assay was also performed after incubation with AuNPs and BC spheres (Figure 4). In this model, the differences at the lowest concentration were observed, but they were higher than the 2D model. The highest peak reached 170% viability after 72 h of incubation with the 30 nm AuNPs. Some similarities between 2D and 3D models were observed. The 48 h TP again became the most toxic for cells using RGD-PEG_2000_-AuNPs_10nm_. Moreover, the mean viability level in each TP and kind of AuNP was again 80–90%. Analyzing the viability level from each type of AuNP alone, some stability trends were observed. Still, comparing AuNPs of all sizes and functionalization, some differences were noticed between those stabilities, especially at the 48 and 72 h TPs. Considering the differences in the distribution level of the AuNPs because of the used models, a diversity in results was observed. The incorporation in the 3D model was uncontrolled, thus, greater differences in viability level were detected. The viability results of the 2D (Table 1) and 3D (Table 2) models were collected as a resume of values for each condition tested. 

### 3.4. Flow Cytometry Analysis

#### 3.4.1. The Reactive Oxygen Species Generation and Time Optimization

Based on the viability level, three concentrations of AuNPs were chosen for further investigation: 0.0004, 0.0008, and 0.0012 mg/mL. In this study, the crucial aim was to check the various conditions of the planned experiments related to the concentration, size, and functionalization of the AuNPs and the incubation time. At first, a wide range of TPs was tested in a 2D model to observe which areas changed in terms of the ROS generation that occurred. The experiment was performed singly at the TPs of 30 min, 2, 4, and 6 h (Figure 5A–D). The increase that was detected at the 30 min TP reached about a three times higher level of ROS compared to the control group. Thus, in the next step, shorter incubation times were investigated. The range of 1 h was investigated, hence, the ROS level was checked every 10 min (Figure 5E–H). The highest level of ROS induction was reached using a 0.0008 mg/mL concentration in all kinds of AuNPs. Only, in the case of RGD-PEG_800_-AuNPs_30nm_, the same results were observed for the smaller and middle-used concentrations. Analyzing shorter periods, a strong trend in the increasing ROS level was detected at the TP of 50 min. This case was recognized in all AuNPs. Due to this, the 30, 50 min, 2, and 4 h TPs of incubation with AuNPs were chosen for further analysis and comparison between the 2D and 3D models. 

To compare the 2D and 3D models, the same concentrations of AuNPs were used. In the 2D model (Figure 6A–D), the same concentration of 0.0008 mg/mL appeared to be the most effective in the ROS induction, in which an about three times higher ROS level was observed compared to the control group. The effect at 30 min was similar to that at 50 min in most cases, but the differences between these two TPs increased after incubation with 10 nm AuNPs (Figure 6C,D). These might be due to dependency—the smaller the size of the AuNP, the deeper the incorporation into the cells. As in the previous preliminary experiments, no effect was detected at 2 and 4 h of incubation. Consequently, the TP of 50 min and 0.0008 mg/mL was chosen as the most effective in generating ROS in the 2D model using all nanostructures. 

More homogenous results were presented in the 3D model (Figure 6E–H). In all kinds of AuNPs, the increase at the 30 min TP was observed. There was a considerable similarity between effects taking into consideration the used concentration of AuNPs, so it was not possible to choose the most effective one. Comparing the results of the AuNPs of different sizes, a diversification was noticed. The bigger AuNPs caused a higher ROS level at the 30 min TP, reaching about 3.5–4.5 higher values in contrast to the 10 nm AuNPs, where the effect was about 1.5–3.0 times higher compared to the control.

Moreover, when analyzing the diversity in PEG length inside the size of the AuNPs, one trend observed was that the higher the ROS level, the shorter the PEG chain. It occurred in both sizes of AuNPs in the 3D model. At 2 and 4 h TPs, the same situation was discovered; no effect of ROS generation was observed. 

#### 3.4.2. Uptake of AuNPs

Based on previous work [17], the evaluation of AuNP uptake was established. The experiment assumed that the higher granularity of the cells, the more nanoparticles they incorporated. This parameter was checked by SSC-H analysis using flow cytometry; higher side scatter levels inform on the higher granularity of the cells. For this analysis, the same concentration and TPs were used as in the experiments for the ROS induction level. After incubation with RGD-PEG_800_-AuNPs_10nm_, RGD-PEG_2000_-AuNPs_10nm_, RGD-PEG_800_-AuNPs_30nm_, and RGD-PEG_2000_-AuNPs_30nm_, the cells were collected and analyzed using the SSC-H parameter in both cell culture models. The results appeared to be highly heterogeneous. The SSC-H level was too low to assess the uptake of cells in the 2D model (Figure 7A–D) and 3D model (Figure 7E–H). It is likely that this is correlated to the size of the cells and used AuNPs. Considering the level of the standard deviations and SSC-H values, no apparent effect was observed. Only, in the case of the 3D model, the smallest size and PEG chain AuNPs, RGD-PEG_800_-AuNPs_10nm_ changed the granularity of MCF-7 cells, reaching 1.7 of the relative SSC-H level. Based on these results, there was no information about the TP or concentration changing the granularity of the cells, however, this is not identical to the situation in that the cells did not absorb the AuNPs. 

### 3.5. Irradiation

#### 3.5.1. Clonogenic Assay

An important part of this research was related to the effect of induction radiosensitivity by AuNPs of MCF-7 cells and comparing those results between the 2D and 3D models. Based on previous optimization in time and AuNP concentration, the TP of 50 min and 0.0008 mg/mL of AuNPs was used for the irradiation experiments. Although the highest level of ROS was detected at 30 min in the 3D model and in the 2D model at 50 min, the second TP was chosen. There was no intention that the irradiation should be implemented in the highest level of ROS, but after ROS induction and thus, the molecular effect they caused. Due to this, the longer TP was used in both models and the middle concentration, which was the most effective in generating ROS, especially in the 2D model. In contrast, similar results were observed in all AuNP concentrations in the 3D model. Cells were irradiated using the dose of 2 Gy after 50 min incubation with RGD-PEG_800_-AuNPs_10nm_, RGD-PEG_2000_-AuNPs_10nm_, RGD-PEG_800_-AuNPs_30nm_, and RGD-PEG_2000_-AuNPs_30nm_ in the concentration of 0.0008 mg/mL. 

Many statistically significant differences were discovered in the 2D model (Figure 8A). The surviving fraction (SF) of 2 Gy was compared to the SF of 2 Gy in addition to the previous incubation with AuNPs. The highly significant differences occurred between SF of 2 Gy and SF of 2 Gy + RGD-PEG_2000_-AuNPs_30nm_ (*p* = 0.000275) and 2 Gy + RGD-PEG_2000_-AuNPs_10nm_ (*p* = 0.00016). Moreover, there were also statistically significant differences between SF of 2 Gy and SF of 2 Gy + RGD-PEG_800_-AuNPs_30nm_ (*p* = 0.018624) and 2 Gy + RGD-PEG_800_-AuNPs_10nm_ (*p* = 0.025877). In this case, there might be a correlation between the used PEG chain because higher statistical differences appeared between previously non-treated and treaded cells with AuNPs functionalized with PEG of 2000 MW.

Moreover, there were statistically significant differences in analyzing the PEG chains or sizes between AuNPs. Considering AuNPs at the size of 30 nm used for previous incubation with cells before irradiation, a statistical difference was observed (*p* = 0.036042) where the SF of 2 Gy + RGD-PEG_2000_-AuNPs_30nm_ compared to the SF of 2 Gy + RGD-PEG_800_-AuNPs_30nm_ decreased. A greater difference was also detected between the SF value of 2 Gy + 2 Gy + RGD-PEG_2000_-AuNPs_10nm_ SF and 2 Gy + RGD-PEG_800_-AuNPs_30nm_ (*p* = 0.002943). Another exciting occurrence was between the smallest and biggest AuNPs. The SF of 2 Gy + RGD-PEG_2000_-AuNPs_30nm_ was lower than that of 2 Gy + RGD-PEG_800_-AuNPs_10nm_ (*p* = 0.025877). The last difference in SF was between AuNPs at the size of 10 nm used before irradiation. Again, AuNPs with the longer PEG chain showed that the SF value decreased using the previously incubated AuNPs functionalized with the longer chain (*p* = 0.002215). The 3D model is inherently more resistant to any exposure. In this model, there was no statistically significant difference between the SF of 2 Gy and SF of 2 Gy + AuNPs. The statistical analysis aiming to compare all variants showed one difference between the cells irradiated and previously incubated with the same size of AuNPs of 30 nm, but with diversity in the PEG chain (*p* = 0.03396). Concerning the objective of this study, this difference had no investigational sense because the SF of 2 Gy + RGD-PEG_800_-AuNPs_30nm_ was higher than the SF value for 2 Gy alone (Figure 8B). 

#### 3.5.2. γH2AX Analysis—Flow Cytometry and Immunofluorescence

After incubation with AuNPs and irradiation with 2 Gy, the γH2AX analysis was performed. At first, using flow cytometry, the γH2AX relative level was checked. The results after treating cells with the previous incubation with AuNPs and 2 Gy of irradiation were compared to the results of 2 Gy alone. Considering the γH2AX relative level in the 2D model (Figure 9A) between variants of 2 Gy and 2 Gy + AuNPs, the most significant difference was between 2 Gy and 2 Gy + RGD-PEG_2000_-AuNPs_30nm_ (*p* = 0.000501). Moreover, similar statistical differences compared the γH2AX relative level of 2 Gy to the γH2AX relative level of 2 Gy + RGD-PEG_800_-AuNPs_10nm_ (*p* = 0.02105) and 2 Gy + RGD-PEG_2000_-AuNPs_10nm_ (*p* = 0.015123). As in the case of SF, there was a difference in the γH2AX relative level between cells treated with 2 Gy and the same AuNP size of 30 nm. Using 30 nm AuNPs with a longer PEG chain turned out to be more effective in the induction of γH2AX compared to the AuNPs with a shorter PEG chain (*p* = 0.020192). In the 3D model (Figure 9B), a similar tendency in results was observed, whereas only one difference was statistically significant. This occurred between the γH2AX relative level after the exposure of cells to 2 Gy alone and 2 Gy + RGD-PEG_2000_-AuNPs_30nm_ (*p* = 0.013027). In other cases, the trend of increasing γH2AX relative level was noticed, but it was not enough for a significant difference. 

The immunofluorescence of γH2AX was investigated (Figure 10 and Figure 11). After the previous 50 min incubation time of cells with AuNPs, the irradiation in the dose of 2 Gy was implemented. Using the image deconvolution, the quantification of the γH2AX foci mean fluorescence intensity (MFI) was performed (Figure 12). The increase in the γH2AX foci MFI was observed in the MCF-7 model where the RGD-PEG_2000_-AuNPs_30nm_ were added. In the other variants, a decrease in the γH2AX foci MFI was detected when comparing both the 2D and 3D models, although lower values were noticed using MCF-7 spheres. 

## 4. Discussion

Introducing the functionalized AuNPs into the tumor area before IR could change the treatment outcome. It is crucial to obtain a high efficiency of their active or passive intracellular transport and the results they could cause in combination with radiotherapy. The radiosensitivity effect of AuNPs is known, but there is a knowledge gap in the case of choosing the most effective experimental conditions related to not only the size of the kind of functionalization, but also to the time and concentration dependent manner of the implementation of AuNPs. The main aim of the study was to assess how the radiosensitivity effect of AuNPs depends on the diversity of the properties of AuNPs and how the effects of different AuNP properties acted in 2D and 3D models. 

AuNPs were synthesized, with the result of 10 mm and 30 nm sized AuNPs and then functionalized with PEG and the RGD peptide. PEG is a polyether compound derived from petroleum with applications in biomedicine [18]. This kind of AuNP functionalization is biocompatible and can protect gold surfaces from aggregation in vitro [19]. Moreover, various types of peptides can be used to allow biomaterials to cross cell membranes, aiming for cancer cell targeting [20]. When examining the penetration efficiency of AuNPs, the different lengths of the attached PEG chain should be taken into account. It was observed [21] that a longer PEG chain resulted in a more significant reduction in nanoparticle toxicity, but at the same time, limited internalization due to increased nanostructure size. Achieving better intracellular transport and avoiding degradation in cell structures is a significant challenge and a rarely undertaken research topic. According to earlier studies, the in vivo blood, circulation time, and clearance rate of AuNPs are reportedly impacted by their surface functionalization. For instance, Cho et al. [22] (examined the biodistribution of PEG-encapsulated AuNPs in tumor-bearing mice. According to their research, within 7 days of intravenous injection, AuNPs (4 and 13 nm) were dispersed throughout the organs including the liver and spleen. Their concentration peaked in these organs, named the reticuloendothelial system (RES) organs. RES organs were disseminated by 100 nm PEG-AuNPs, which peaked 30 min after intravenous injection and remained at high levels for six months. This discovery is supported by numerous investigations that have demonstrated that greater PEG chain lengths aided in the longer blood circulation times oof NPs [23].

Another important aspect is the method of evaluating the effectiveness of nanoparticles depending on the research model used. Promising results obtained in 2D conditions are often not reflected in the results of in vivo studies using animal models. 2D cultures are devoid of a 3D network of different cell types surrounded by an extracellular matrix. These two aspects play pivotal roles in the diffusion and uptake of nanoparticles by cells. However, the evaluation of their performance remains limited, as most work is carried out on 2D culture systems [24]. To extend the existing reports, research using 3D systems should be implemented. Spheroids can be formed by single cells that can form cell aggregates. During proliferation, intercellular communication is established, and cells form a specific microenvironment. The findings suggest that mammosphere culture of metastatic cells could be an excellent model for studying the sensitivity of tumorigenic stem cells to therapeutic agents and characterizing the tumor-inducing subpopulation of breast cancer cells [25]. Yousefnia et al. [26] demonstrated the stemness phenotypes of mammospheres generated for further applications in therapeutic approaches and used the MCF-7, MDA-MB-231, and SKBR3 BC cell lines. MCF-7 had the highest mammosphere formation efficiency. Mammospheres generated from all three cell lines had significantly higher proliferation, migration rate, and drug resistance. Moreover, the deposition of the extracellular matrix prompts spheroids to study the ability of nanoparticles to penetrate and diffuse in solid tumors. 

At first, the toxicity of the AuNPs was evaluated. The viability of cells slightly decreased after 24, 48, and 72 h compared to the control, and reached the lowest AuNP concentrations of about 90%. Based on these results, the lowest three concentrations were chosen for the following experiments, aiming to define the granularity of the cells and ROS level after incubation with AuNPs. Analysis of the results in the 2D and 3D models at a concentration of 0.002 mg/ML indicated heterogenous viability using different kinds of AUNPs. Concentrations in the range of 0.002–0.02 mg/ML in the 2D model induced viability at 80%, whereas in the 3D model, viability was more varied, indicating that the AuNP distribution is related to different cell morphology. Interestingly, in 3D culture, the viability seemed to increase at the lowest concentration of 30 nm AuNPs, reaching about 170% of the viability. In fact, some studies that investigated the biological effects of nanoparticles found a hormetic dose-response [27]. As a result, it appears clear that future research should focus on this issue by investigating the potential adverse health effects caused by low-level nanoparticle exposure [28]. Xia et al. [29] investigated the toxicity levels of 5 nm AuNPs in the HepG2 cancer cells. They suggested that smaller AuNPs are more likely to accumulate in the cell after entering the cytoplasm and were confined within the endosome or lysosome. Acidic pH in the endosomes and lysosomes may induce the release of toxic ions, indicating ion-specific toxicity. Rostami et al. [30] demonstrated that MCF-7 cells treated with glucose-coated AuNPs decreased the cell viability by 13.2%. 

One of the main mechanisms claimed to be the potential reason for radiosensitivity is ROS induction after AuNPs are introduced to the BC cells. It is the one widely recognized biological pathway of radiosensitization. ROS may generate cell damage directly by interacting with biomolecules including cellular DNA or cause cell death indirectly by the oxidation of lipids, proteins, and mitochondrial dysfunction [31]. In this study, to implement the time–concentration manner of using AuNPs, three AuNP concentrations of 0.0004, 0.0008, and 0.0012 mg/mL were used at four TPs of 30 min, 2, 4, and 6 h. The enhancement in ROS generation was noticed at 30 min of incubation. Due to this, the ROS level was analyzed in shorter periods, from point “0” to 60 min per every 10 min in the real-time analysis. The most effective concentration, causing the highest ROS level, was 0.0008 mg/mL at 50 min of the cell incubation with AuNPs. The next step was implementing the optimized results to compare the 2D and 3D models. The same concentrations of AuNPs were used, but the TPs were 30 min, 50 min, 2, and 4 h. The main difference between the models was that in the 3D model, the highest point of ROS level was reached at 30 min, in contrast to the 2D model where the ROS induction was similar at 30 min and 50 min, but the trend of highest ROS peak was at 50 min of the experiment. As in the optimization, performed using a 2D model, the increasing ROS level was noticed with 0.0008 mg/mL of AuNPs. In the 3D model, all concentrations achieved similar results of ROS level enhancement. Considering the 2D and 3D models, differences in ROS generation between results gained with the 30 nm AuNPs were observed. Values in the 3D model were significantly higher than the quantification detected in the 2D model. Analyzing the PEG role in the ROS induction, in the 2D model, there was no indication of a difference between the AuNPs whereas in the 3D model, the enhancement of ROS generation was noticed using AuNPs with the shorter PEG chain. This suggests that ROS induction is associated more with the presence of AuNPs, but not with the used surface modification. In addition, the enhancement of ROS in a short treatment time may lead to a cytotoxic effect due to the triggering of oncogenic signaling pathways by ROS [32]. Kowsalya et al. [33] measured the intracellular ROS production in MCF-7 cells after incubation with AuNPs. They indicated that ROS production increased significantly after using a higher concentration of AuNPs (40.63 +/− 1.69 µg/mL). This did not correlate with our results, which might be associated with another synthesis method. They synthesized AuNPs using green synthesis. Moreover, the AuNPs in our study were functionalized to increase the uptake, thus a lower AuNP concentration was needed for the induction of ROS. 

Furthermore, AuNPs altered the cellular redox state in the MCF-7 cells by generating intracellular ROS, thereby inducing ROS-mediated cell death. Another group [34] investigated whether the chitosan functionalized AuNPs, in sizes of 3–10 nm, could induce ROS production in MCF-7 and HeLa cells. The level of ROS generation similarly reached 35% in both cell lines compared to the control group. 

In our previous work [17] and the article published by Wu et al. [35], the granularity of cells was analyzed. The uptake after the cells’ incubation with AuNPs can be established using the SSC-H signal, as the presence of AuNPs raised the SSC-H response. Using the AuNP functionalization of PEG and RGD, we tried to enhance the AuNP internalization process. Functionalization with targeting biomolecules highly induces the tracking behavior and indicates the localization of AuNP accumulation. In our study, pegylated AuNPs were conjugated with Arg–Gly–Asp peptides to target BC cells. It is usually used for targeting cells expressing RGD-binding integrins such as αV-integrins [36]. The SSC-H parameter indicating the cellular uptake did not alter enough during selected TPs to indicate changing granularity. The probable reason could be based on the size of the AuNP cells and MCF-7 cell morphology, as the 10 and 30 nm AuNPs may be too small to generate differences in the SSC-H parameter. Moreover, the value of the MCF-7 cells could reach about 3375 to 16,873 µm^3^, which makes them one of the bigger cells, compared to, for example, MCF10A cells [37]. Gaining the highest uptake might be undetectable in this kind of cell. Despite the knowledge regarding a diverse morphology of cells in different culture models, there were no statistical differences between the 2D and 3D models analyzing the SSC-H parameter. Since the evaluation of uptake was not conclusive in this experiment, the following step could be functionalized AuNPs with a fluorescent probe to better detect their localization as well as to perform experiments with chemical inhibitors of endocytosis to assess the molecular mechanisms. The diffusion of AuNPs into the cancerous spheres plays a crucial role in nanostructure optimization as it promotes the ability of AuNPs to penetrate and enter the hypoxic and necrotic cores of tumors [38]. The researchers in [39] studied 15 nm AuNPs and 50 nm AuNPs in the CAL-27 and HeLa cell lines using the 2D and 3D culture models. The results of the depth of the penetration tests, an essential property that enables the targeting of deep-set tumor cells, showed that 15 nm AuNPs penetrated more successfully than the 50 nm AuNPs. 

The main aim of the study was to assess the radiosensitivity effect of MCF-7 cells after irradiation and the previous incubation with AuNPs as well as its dependence on the properties of the AuNPs, and a comparison of the obtained results between the 2D and 3D models. Radiotherapy is an irreplaceable treatment for the effective control of local tumors, mostly connected with chemotherapy and surgical therapy [38]. Numerous studies have shown that using nanoplatforms to deliver therapeutic compounds to tumor tissues in a targeted manner increases the bioavailability of cytotoxic medicines while reducing the risk of damage to healthy tissues. Additionally, research has indicated that one of the most crucial elements impacting radiation effectiveness is the biodistribution of AuNPs [39]. They cause high local ionization in tumor tissues, which minimizes the treatment time and radiation doses, providing that they have been dispersed preferentially in the tumor area and it follows the same energy absorption mode as that of the surrounding healthy tissues. 

Based on the viability, ROS, and uptake tests, the point of implementing the IR in the dose of 2 Gy was after 50 min of incubation with 0.0008 mg/mL of all kinds of AuNPs. Using the clonogenic assay, the SF was analyzed by comparing the SF of the cells irradiated with 2 Gy alone to cells previously incubated with AuNPs and irradiated with an IR dose. In the 2D model, statically significant differences were noticed. AuNPs combined with 2 Gy caused a significant SF reduction. There was a similar tendency to analyze values inside the AuNP sizes. In each case, AuNPs conjugated with PEG in MW of 2000 showed a decreased SF value compared to PEG_800_. This probably suggests that scientists should consider longer PEG chains during the AuNP decoration. Results established in the 3D models were not so promising. There was no significant difference between the cells treated with the AuNPs and IR combination and the cells irradiated alone, although the lowest results were caused using RGD-PEG_2000_-AuNPs_30nm_. The dose of 2 Gy was chosen because the aim was to apply the scheme of a patient treatment. In clinical conditions, the administration of AuNPs could be performed daily as the radiotherapy fractions, but it depends highly on the time distribution of AuNPs in the blood circulation. The observed radiosensitivity effect might also be caused by ROS generation after incubation with AuNPs. The “energy powerhouse of cells” and amplifiers of ROS generation are mitochondria and is one of the crucial targets that must not be overlooked during the radiosensitization process [40]. Apoptosis may result from oxidative stress-induced mitochondrial malfunction, which includes damage to mitochondrial DNA (mtDNA), the mobilization of cytochrome C, and other biological repercussions [41]. Additionally, it has been proposed that AuNP-induced oxidative stress may be caused by the suppression of proteins necessary for cellular oxidative homeostasis such as thioredoxin reductase TrxR1 [42]. Further research is required to fully understand the mechanism of AuNP-induced oxidative stress, which is currently thought to originate from mitochondrial malfunction brought on by high intracellular ROS levels. 

The nanoparticles used in this study may be employed to deliver bioactive compounds to enhance the therapeutic effect of ionizing radiation. To minimize the side effects and increase the effectiveness of the therapy, one proposal is use a drug delivery system containing gold nanoparticles incorporated into liposomes with bioactive substance or as a factor changing the biological response of cells to ionizing radiation. The combination of nanoparticles with a new delivery system could increase the internalization of the nanomaterial, especially in the 3D model, and the bioactive substance would improve the response of cancer cells to combined radiotherapy. According to some research results [43], AuNPs decreased the proliferation, migration, and invasion processes of BCPAP and TPC-1 papillary thyroid cancer cells. The group also showed that AuNPs reduced CCT3 mRNA expression in the papillary thyroid cancer cells, which further demonstrated their antitumor effects. These discoveries might result in the creation of a new potential method for combating various cancer types. 

Comparison of 2D and 3D cell culture models by Fontoura et al. [10] indicated that tumor cells grown on 3D models are more resistant to chemotherapy drugs and have similar morphologies resembling in vivo tumors. This could be related to our results that the 3D model reacted differently than the 2D model. The clinical promise of this novel therapy was further supported by a study by Yang et al. [44], who showed the therapeutic benefit of the combination therapy on MDA-MB-231 cells in the presence of MV radiation using RGD-modified AuNPs in conjunction with the chemotherapy medication cisplatin. The survival rate of cells treated with AuNP-RGD:Cis and radiation was reduced noticeably, even at low concentrations, at values substantially lower than the survival rate of cells treated with cisplatin and radiation alone. As a result, they anticipate that this AuNP-mediated chemoradiation will soon be incorporated into the treatment of cancer. Raitanen et al. [45] published an article describing the radiation response comparison in 2D and 3D cell cultures of various cancer cell lines (PC-3, LNCaP, and T-47D) irradiated with 1, 2, 4, 6, 8, or 20 Gy doses of X-ray beams. The findings support the existence of significant diversity in radiobiological response to X-rays in the 2D and 3D cell culture models. They used the same 3D cell culture method as in our study. The results showed that the radiobiological response to X-rays measured in 2D was not reflected in 3D. When compared to the respective monolayers, the spheroid model demonstrated higher radioresistance in all cell lines. To evaluate the radiosensitization effect of AuNPs, Hebert et al. [46] created 5 nm AuNPs coated with the gadolinium chelating agent dithiolated diethylenetriaminepentaacetic gadolinium (Au@DTDTPA:Gd) for the in vitro and in tumor-bearing mice (MC7-L1) analysis. Although AU@DTDPA:Gd indicated no radiosensitizing effect in vivo, in vitro experiments revealed that cell death was induced at 5 mM, a concentration 100 times lower than that detected in the tumors. One proposed reason for this toxicity was the free Gd distribution in the tumor as a result of the surface instability of the AuNPs. Despite promising preclinical confirmation in vitro and a small number of in vivo experiments as well as approaches in radiobiological understanding, AuNPs have not yet entered the clinic. This could be due to a mismatch between the expected levels of radiosensitization based on the experimental conditions and analyzed radiobiological response as well as a limited mechanistic explanation [47]. When designing experiments, many factors can be taken into account to accelerate the introduction of AuNPs into clinical treatment. The use of megavolt energy as well as cell lines isolated from patients are one of many examples that will produce nanoparticles used in the treatment of patients in the near future. 

Single-strand breaks (SSBs), double-strand breaks (DSBs), DNA-protein cross-links, and DNA base alterations are only a few types of DNA damage that radiotherapy can cause. Among them, DSBs are the most destructive kind of radiation-induced damage and are connected to the destruction of clonogenic cells. Cell death can occur in a number of ways if the genomic stability is compromised by the inability to repair DNA DSBs. The earliest sensitive markers are thought to be p53-binding protein 1 (53BP1) and phosphorylated histone variant γH2AX [48]. In our study, γH2AX, after exposure of the cells to a dose of 2 Gy and a dose of 2 Gy with the previous 50 min incubation with AuNPs was analyzed. As in the previous clonogenic results, the results established in the 2D model presented higher statistically significant differences. There was a noticeable increase in every relative γH2AX level in the cells treated previously with AuNPs. The highest value was reached using RGD-PEG_2000_-AuNPs_30nm_, again indicating that those AuNPs are the most effective in inducing the radiosensitivity effect of BC cells in both models. Based on the quantification results of the γH2AX foci MFI, RGD-PEG_2000_-AuNPs_30nm_ induced the increase compared to the control group (2 Gy alone). On one hand, this result is associated with the flow cytometry outcome. On the other hand, analyzing the rest of the values, the opposite results were detected. Using flow cytometry, in all variants, the γH2AX relative level increased, whereas using the γH2AX foci MFI based on the immunofluorescence, a reduction was observed compared to cells irradiated with 2 Gy alone. This might be related to the number of cell used in both analysis; for flow cytometry, 200,000 cells were used while for immunofluorescence quantification, it was a fragment (image) of 30,000 cells used in the staining. In addition, the increase in the γH2AX relative level correlates with the obtained SF value. Therefore, DNA repair inhibition appears to be a crucial physiologic process underlying AuNP radiosensitization. However, other researchers contend that AuNPs have little impact on the kinetics of DNA repair [49].

## 5. Conclusions

Implementing gold nanoparticles into the tumor area is a potential strategy to solve the problem of protecting healthy tissues in the beam field. Sensitizing tumor cells to radiotherapy would allow for lower radiation doses, reducing the negative effects of radiation on healthy tissues. This involves still unknown physical, chemical, and biological processes. The results of this research suggest that scientists might pay more attention to the conditions when planning experiments depending on the effects they wish to obtain. 

The size and functionalization of nanoparticles have a very significant impact on the way they penetrate the cell. The study highlighted the role of the PEG chain in the efficiency of the AuNPs in the process of sensitizing cells to ionizing radiation. In both sizes of AuNPs, the 2000 MW PEG led to greater statistically significant differences than the AuNPs functionalized with a PEG of 800 MW when analyzing the 2D model. The most effective AuNPs in inducing the sensitivity for IR were the biggest in size and functionalization AuNPs used in both cell culture models. 

In the future, scientists should consider a time-dependent manner using AuNP planning experiments, aiming to decrease SF after exposure to AuNPs and IR as well as the size of PEG in functionalization. Results obtained in the 2D model were satisfactory enough to claim that AuNPs are the appropriate solution for combined treatment with RT. In the 3D model, the results after IR suggest that further research is needed to gain a higher efficiency of AuNPs in the sensitizing of BC cells. The future clinical success of the use of nanoparticles is predicted based on a more detailed understanding of the mechanisms by which their physicochemical properties influence the cellular radiobiological response. 

## Figures and Tables

**Figure 1 pharmaceutics-15-00862-f001:**
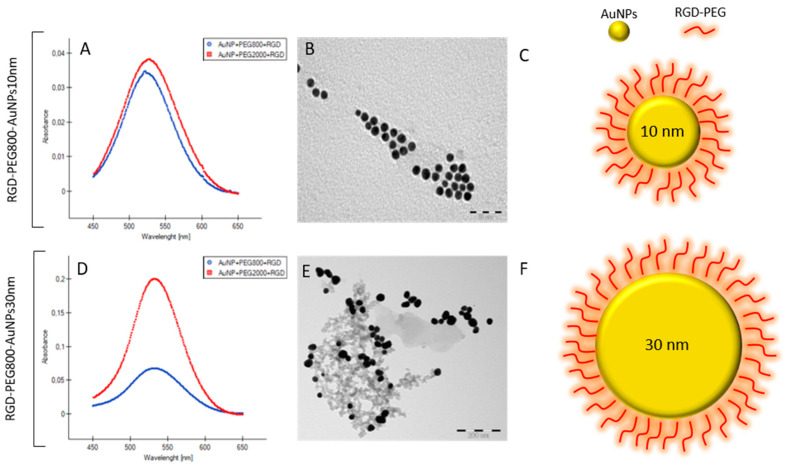
Characterization and morphology of the synthesized AuNPs. (**A**) UV–VIS spectrum for RGD-PEG_800_-AuNPs_10nm_ and RGD-PEG_2000_-AuNPs_10nm_. (**B**) TEM image of 10 nm AuNPs, based on the TEM measurements the average size of AuNPs was 10.5 nm +/− 1.3 nm. (**C**) The scheme of RGD-PEG_800/2000_-AuNPs_10nm_. (**D**) UV–VIS spectrum for RGD-PEG_800_-AuNPs_30nm_ and RGD-PEG_2000_-AuNPs_30nm_. (**E**) TEM image of 30 nm AuNPs, based on the TEM measurements, the average size of AuNPs was 30.9 nm +/− 1.8 nm. (**F**) The scheme of RGD-PEG_800/2000_-AuNPs_30nm_.

**Figure 2 pharmaceutics-15-00862-f002:**
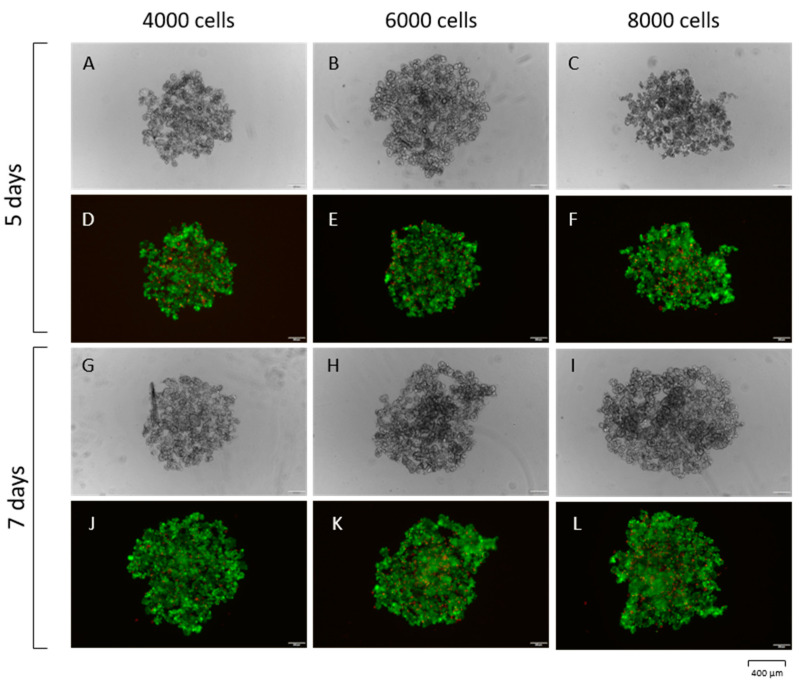
Live/dead assay of the MCF-7 3D model. BC spheres were observed after 5 days of forming in a bright field containing 4000 cells (**A**), 6000 cells (**B**), and 8000 cells (**C**). The identical spheres were tested with the live/dead assay (**D**–**F**). After 7 days of formation, identical BC spheres were photographed using the bright field (**G**–**I**) and live/dead assay (**J**–**L**). The FITC channel was used to present the green alive cells, and the Texas Red channel for the red, dead ones.

**Figure 3 pharmaceutics-15-00862-f003:**
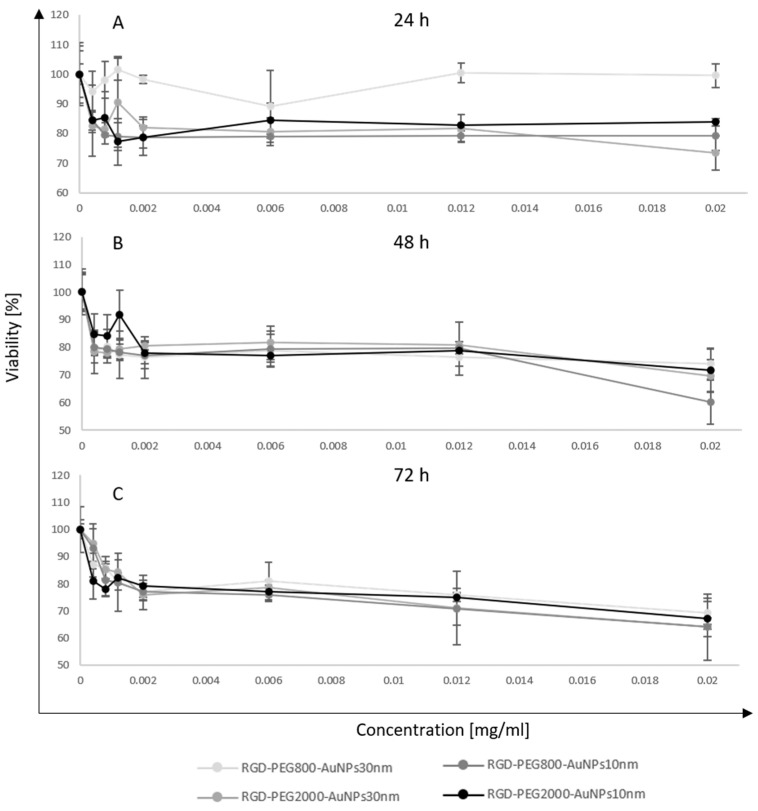
The viability level of the MCF-7 2D model after incubation with AuNPs. In this test, RGD-PEG_800_-AuNPs_10nm_, RGD-PEG_2000_-AuNPs_10nm_, RGD-PEG_800_-AuNPs_30nm_, and RGD-PEG_2000_-AuNPs_30nm_ were used in concentrations of 0.0004, 0.0008, 0.0012, 0.0020, 0.0060, 0.0120 and 0.0200 mg/mL. The time points of incubation time with AuNPs were 24 h (**A**), 48 h (**B**), and 72 h (**C**).

**Figure 4 pharmaceutics-15-00862-f004:**
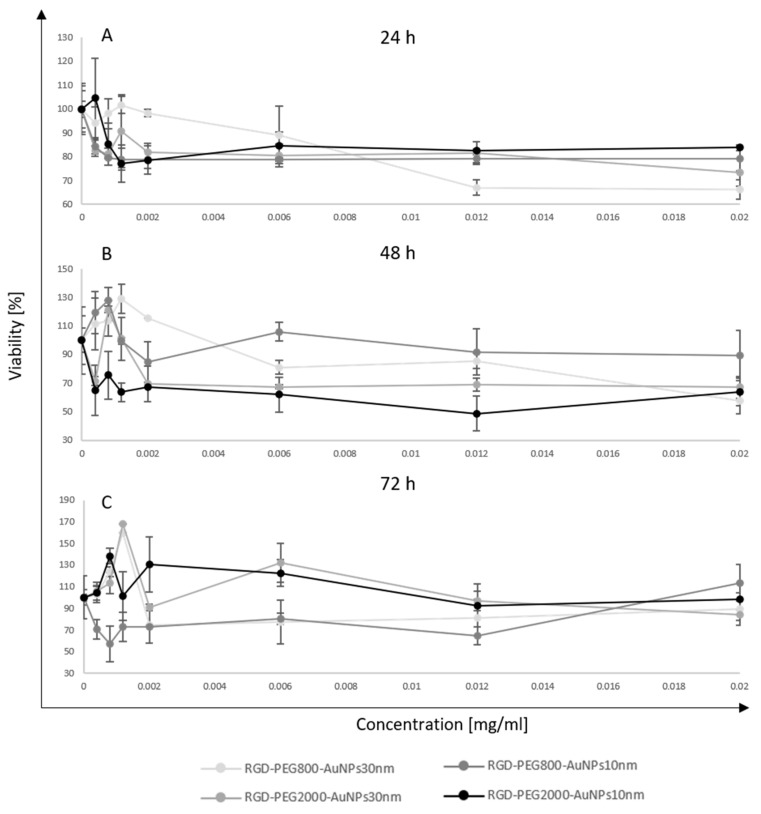
The viability level of the MCF-7 3D model after incubation with AuNPs. In this test, RGD-PEG_800_-AuNPs_10nm_ and RGD-PEG_2000_-AuNPs_10nm_, RGD-PEG_800_-AuNPs_30nm_ and RGD-PEG_2000_-AuNPs_30nm_ were used in concentrations of 0.0004, 0.0008, 0.0012, 0.0020, 0.0060, 0.0120 and 0.0200 mg/mL. The time points of incubation time with AuNPs were 24 h (**A**), 48 h (**B**), and 72 h (**C**).

**Figure 5 pharmaceutics-15-00862-f005:**
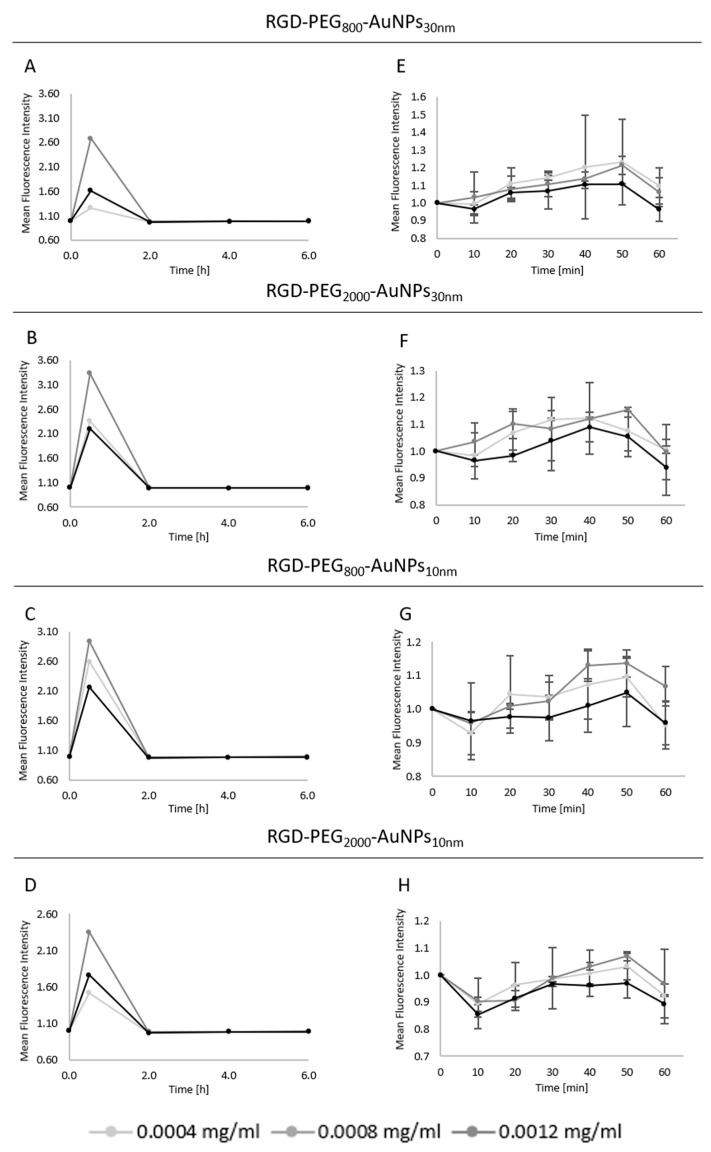
The ROS induction—optimization of the incubation time and AuNP concentration. The preliminary experiments were performed to optimize the TPs of the highest level of ROS generation. The experiment was performed singly ats TP of 30 min, 2, 4, and 6 h incubation with RGD-PEG_800_-AuNPs_10nm_ (**A**), RGD-PEG_2000_-AuNPs_10nm_ (**B**), RGD-PEG_800_-AuNPs_30nm_ (**C**). and RGD-PEG_2000_-AuNPs_30nm_ (**D**). The range of 1 h was investigated; thus, the ROS level was checked every 10 min using RGD-PEG_800_-AuNPs_10nm_ (**E**), RGD-PEG_2000_-AuNPs_10nm_ (**F**), RGD-PEG_800_-AuNPs_30nm_ (**G**), and RGD-PEG_2000_-AuNPs_30nm_ (**H**).

**Figure 6 pharmaceutics-15-00862-f006:**
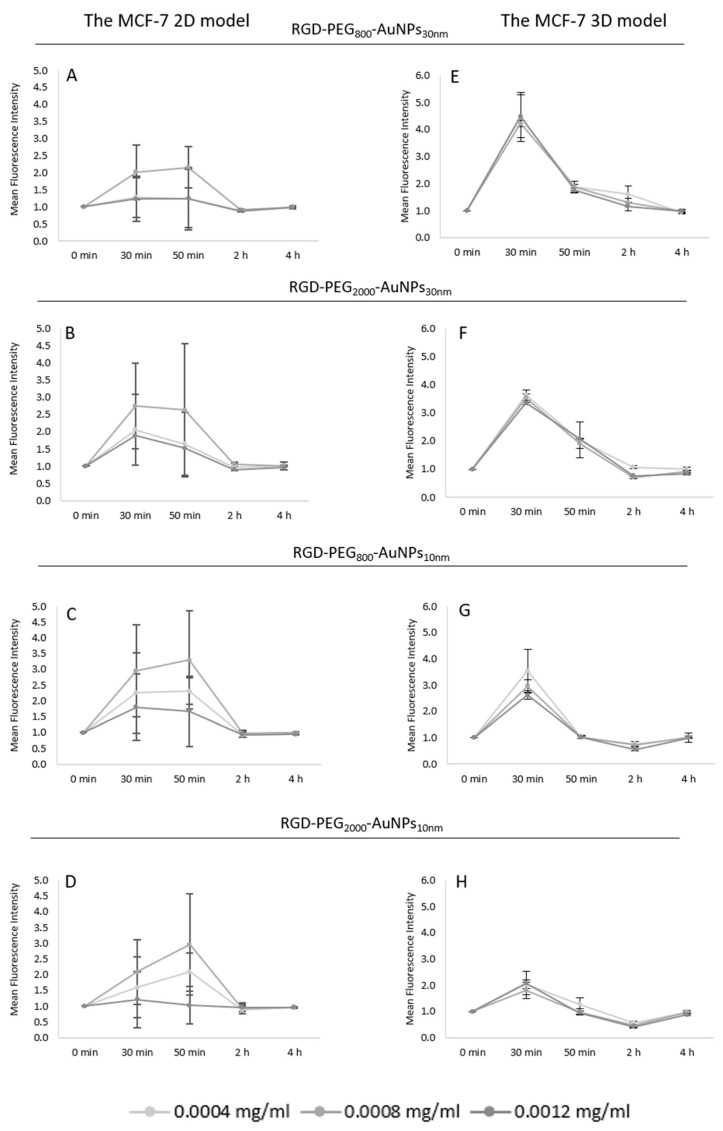
The comparison of the ROS induction level between the 2D and 3D models. In the 2D model, the experiment was performed in triplicate at the TPs of 30, 50 min, 2, and 4 h incubation with RGD-PEG_800_-AuNPs_10nm_ (**A**), RGD-PEG_2000_-AuNPs_10nm_ (**B**), RGD-PEG_800_-AuNPs_30nm_ (**C**), and RGD-PEG_2000_-AuNPs_30nm_ (**D**). In the 3D model, the experiment was performed in triplicate at the TPs of 30, 50 min, 2, and 4 h incubation with RGD-PEG_800_-AuNPs_10nm_ (**E**), RGD-PEG_2000_-AuNPs_10nm_ (**F**), RGD-PEG_800_-AuNPs_30nm_ (**G**), and RGD-PEG_2000_-AuNPs_30nm_ (**H**). The concentrations of all AuNPs were 0.0004, 0.0008, and 0.0012 mg/mL.

**Figure 7 pharmaceutics-15-00862-f007:**
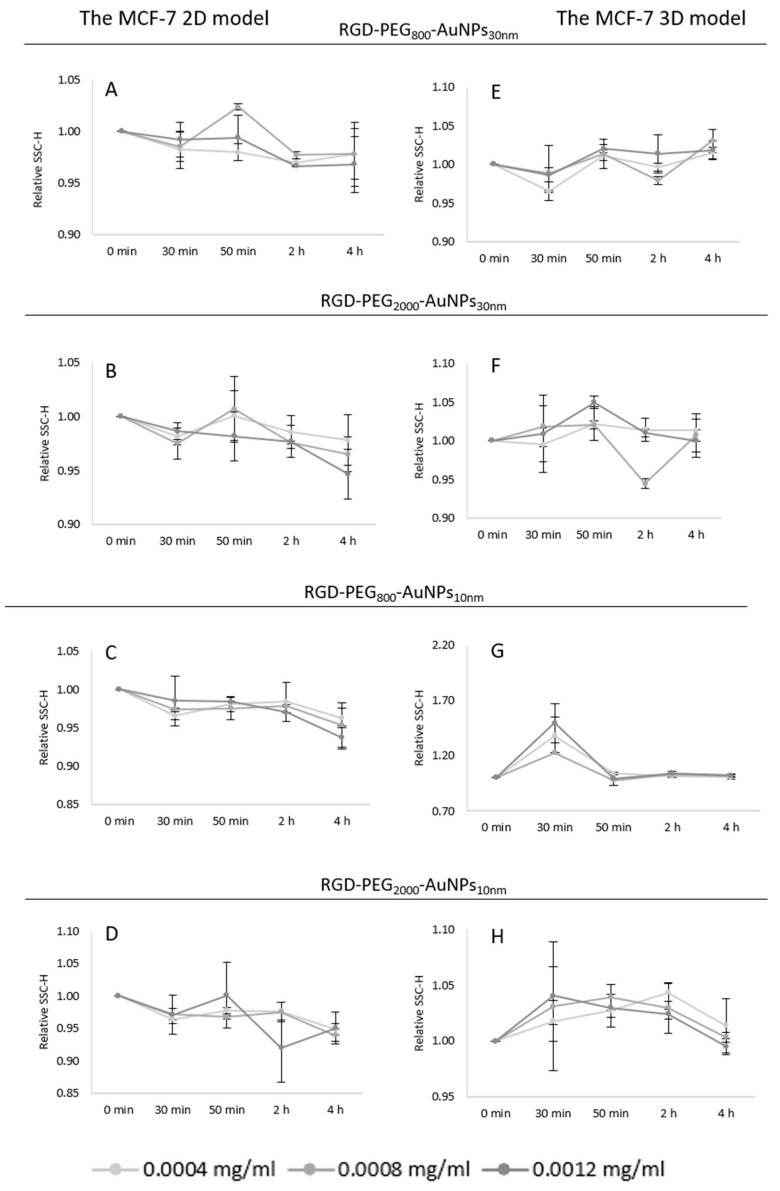
The relative SSC-H parameter level of MCF-7 cells in the 2D and 3D models. In the 2D model, the experiment was performed in triplicate at the TPs of 30, 50 min, 2, and 4 h incubation with RGD-PEG_800_-AuNPs_10nm_ (**A**), RGD-PEG_2000_-AuNPs_10nm_ (**B**), RGD-PEG_800_-AuNPs_30nm_ (**C**), and RGD-PEG_2000_-AuNPs_30nm_ (**D**). In the 3D model, the experiment was performed in triplicate at the TPs of 30, 50 min, 2, and 4 h incubation with RGD-PEG_800_-AuNPs_10nm_ (**E**), RGD-PEG_2000_-AuNPs_10nm_ (**F**), RGD-PEG_800_-AuNPs_30nm_ (**G**), and RGD-PEG_2000_-AuNPs_30nm_ (**H**). All of the concentrations of the AuNPs were 0.0004, 0.0008, and 0.0012 mg/mL.

**Figure 8 pharmaceutics-15-00862-f008:**
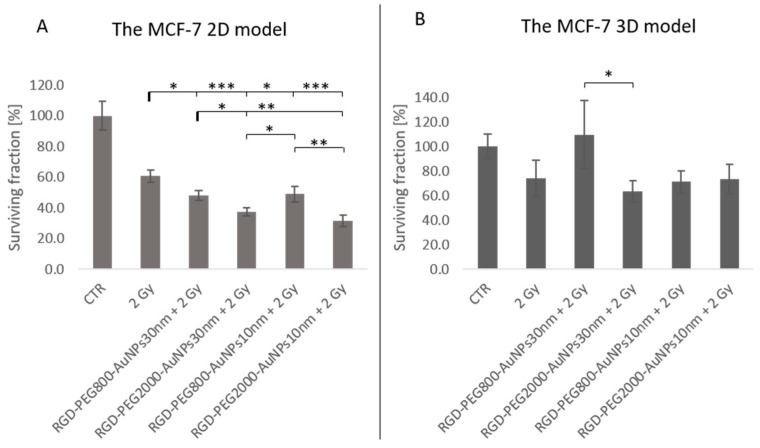
The survival fraction of MCF-7 cells after exposure to a dose of 2 Gy and dose of 2 Gy with the previous 50 min incubation with RGD-PEG_800_-AuNPs_10nm,_ and RGD-PEG_2000_-AuNPs_10nm_, RGD-PEG_800_-AuNPs_30nm_, and RGD-PEG_2000_-AuNPs_30nm_ in the concentration of 0.0008 mg/mL. (**A**) Results for the MCF-7 2D model and (**B**) for the MCF-7 3D model. The setting of the *p*-value * *p* < 0.05, ** *p* < 0.01, *** *p* < 0.001.

**Figure 9 pharmaceutics-15-00862-f009:**
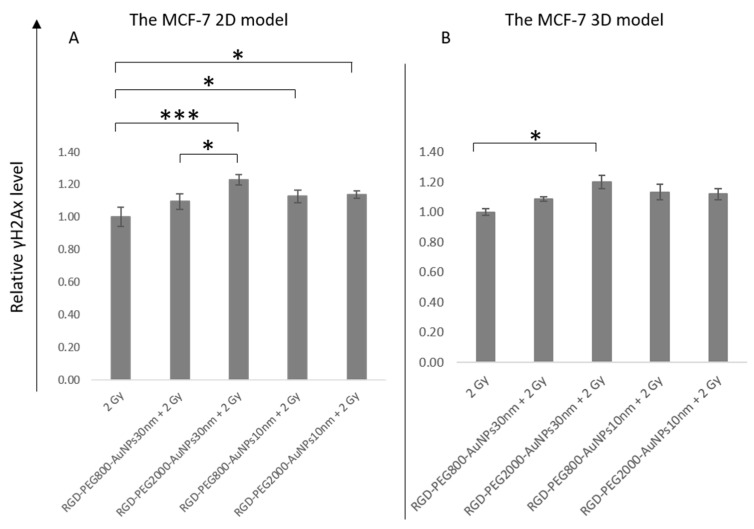
The γH2AX relative level of the MCF-7 cells after exposure to a dose of 2 Gy and dose of 2 Gy with the previous 50 min incubation with RGD-PEG_800_-AuNPs_10nm_, RGD-PEG_2000_-AuNPs_10nm_, RGD-PEG_800_-AuNPs_30nm_, and RGD-PEG_2000_-AuNPs_30nm_ in the concentration of 0.0008 mg/mL. (**A**) Results for the MCF-7 2D model and (**B**) for the MCF-7 3D model. The setting of the *p*-value * *p* < 0.05, *** *p* < 0.001.

**Figure 10 pharmaceutics-15-00862-f010:**
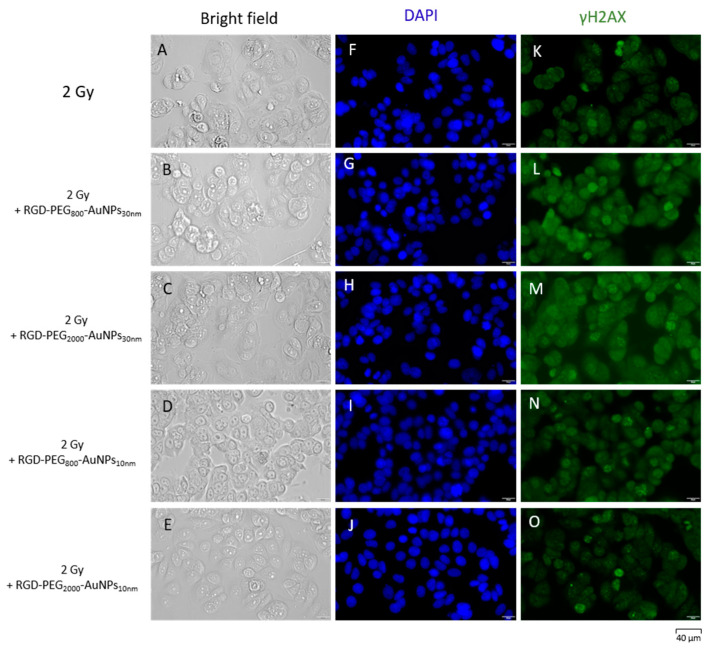
The immunofluorescence of γH2AX after cell exposure to 2 Gy and dose of 2 Gy with the previous 50 min incubation with AuNPs in the 2D model. The morphology of cells was presented in the bright field (**A**–**E**). The nuclei of cells were dyed with the DAPI solution (**F**–**J**). The γH2AX foci were dyed using the solution in green (488 nm) (**K**–**O**).

**Figure 11 pharmaceutics-15-00862-f011:**
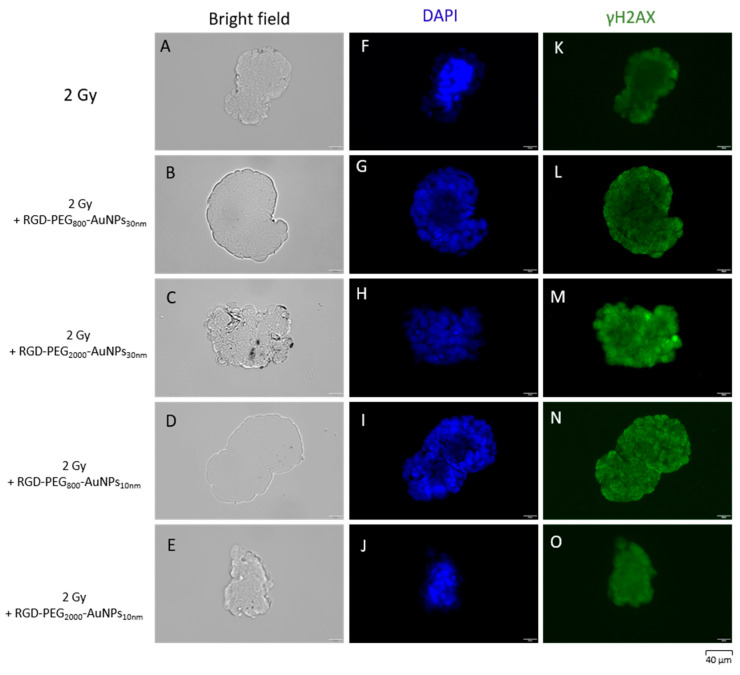
The immunofluorescence of γH2AX after cell exposure to 2 Gy and dose of 2 Gy with the previous 50 min incubation with AuNPs in the 3D model. The morphology of cells was presented in the bright field (**A**–**E**). The nuclei of cells were dyed with DAPI solution (**F**–**J**). The γH2AX foci were dyed using the solution in green (488 nm) (**K**–**O**).

**Figure 12 pharmaceutics-15-00862-f012:**
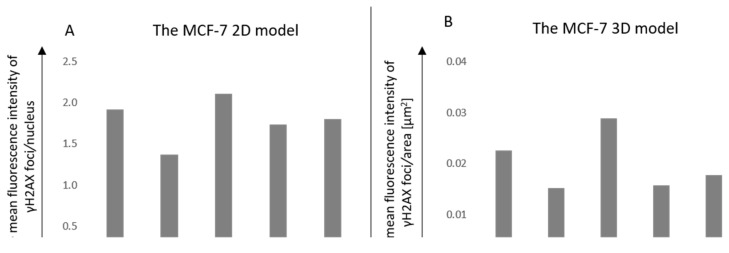
The quantification of the mean fluorescence intensity of the γH2AX foci after cell exposure to 2 Gy and dose of 2 Gy with the previous 50 min incubation with AuNPs in the 2D model (**A**) and 3D model (**B**).

**Table 1 pharmaceutics-15-00862-t001:** The resume of the viability level of the MCF-7 2D model after incubation with AuNPs. In this test, RGD-PEG_800_-AuNPs_10nm_, RGD-PEG_2000_-AuNPs_10nm_, RGD-PEG_800_-AuNPs_30nm_, and RGD-PEG_2000_-AuNPs_30nm_ were used in concentrations of 0.0004, 0.0008, 0.0012, 0.0020, 0.0060, 0.0120, and 0.0200 mg/mL. Abbreviations: SD—standard deviation.

The Cell Viability of the MCF-7 2D Model									
	Concentration mg/mL	RGD-PEG800-AuNPs30nm		RGD-PEG2000-AuNPs30nm		RGD-PEG800-AuNPs10nm		RGD-PEG2000-AuNPs10nm	
		Mean [%]	SD [%]	Mean [%]	SD [%]	Mean [%]	SD [%]	Mean [%]	SD [%]
24 h	0.0000	100.00	9.65	100.00	10.73	100.00	3.42	100.00	7.82
	0.0004	93.95	6.90	82.49	2.23	84.41	3.20	84.37	12.00
	0.0008	98.08	6.20	81.29	2.28	79.51	0.46	85.23	8.90
	0.0012	101.65	3.64	90.63	15.31	78.87	4.59	77.11	7.87
	0.0020	98.30	1.42	81.91	3.58	78.72	3.81	78.62	6.03
	0.0060	89.17	12.16	80.45	4.71	78.95	0.66	84.47	5.75
	0.0120	100.44	3.25	81.56	4.66	79.15	1.80	82.68	0.62
	0.0200	99.48	4.09	73.46	5.78	79.21	5.07	83.79	1.23
48 h	0.0000	100.00	7.13	100.00	6.21	100.00	6.67	66.67	8.23
	0.0004	78.42	7.75	78.65	1.51	80.05	5.68	84.82	7.28
	0.0008	80.33	6.02	77.94	1.79	79.53	0.29	84.20	7.54
	0.0012	77.41	8.56	79.51	3.80	78.19	3.03	91.69	9.00
	0.0020	76.31	7.52	80.63	1.60	77.18	4.82	77.86	3.82
	0.0060	78.80	6.00	81.65	6.14	79.52	6.24	76.95	2.43
	0.0120	76.35	3.15	80.99	0.93	79.59	9.58	78.75	1.15
	0.0200	73.95	5.83	69.71	5.84	60.27	8.07	71.66	7.75
72 h	0.0000	100.00	8.43	100.00	1.96	100.00	3.49	100.00	8.34
	0.0004	86.93	4.36	94.99	7.10	92.92	7.33	80.93	6.42
	0.0008	85.70	4.29	85.16	3.17	81.25	5.93	78.11	2.51
	0.0012	80.80	3.09	84.26	4.42	80.49	10.64	82.18	0.77
	0.0020	77.00	1.94	75.82	5.46	77.06	3.18	79.22	5.24
	0.0060	81.01	7.05	78.62	2.76	75.97	2.35	76.96	1.19
	0.0120	75.80	2.38	71.07	13.47	70.68	6.00	75.09	3.90
	0.0200	69.23	4.21	64.10	12.24	64.07	3.54	67.06	3.70

**Table 2 pharmaceutics-15-00862-t002:** The resume of the viability level of the MCF-7 3D model after incubation with AuNPs. In this test, RGD-PEG_800_-AuNPs_10nm_ and RGD-PEG_2000_-AuNPs_10nm_, RGD-PEG_800_-AuNPs_30nm_ and RGD-PEG_2000_-AuNPs_30nm_ were used in concentrations of 0.0004, 0.0008, 0.0012, 0.0020, 0.0060, 0.0120 and 0.0200 mg/mL. Abbreviations: SD—Standard Deviation.

The Cell Viability of the MCF-7 3D Model									
	Concentration mg/mL	RGD-PEG800-AuNPs30nm		RGD-PEG2000-AuNPs30nm		RGD-PEG800-AuNPs10nm		RGD-PEG2000-AuNPs10nm	
		Mean [%]	SD [%]	Mean [%]	SD [%]	Mean [%]	SD [%]	Mean [%]	SD [%]
24 h	0.0000	100.00	9.65	100.00	10.73	100.00	3.42	100.00	7.82
	0.0004	93.95	6.90	82.49	2.23	84.41	3.20	104.62	16.64
	0.0008	98.08	6.20	81.29	2.28	79.51	0.46	85.23	8.90
	0.0012	101.65	3.64	90.63	15.31	78.87	4.59	77.11	7.87
	0.0020	98.30	1.42	81.91	3.58	78.72	3.81	78.62	6.03
	0.0060	89.17	12.16	80.45	4.71	78.95	0.66	84.47	5.75
	0.0120	66.96	3.25	81.56	4.66	79.15	1.80	82.68	0.62
	0.0200	66.32	4.09	73.46	5.78	79.21	5.07	83.79	1.23
48 h	0.0000	100.00	17.18	100.00	17.08	100.00	23.54	100.00	8.57
	0.0004	111.37	18.37	71.28	3.15	119.50	14.85	64.92	17.77
	0.0008	113.53	10.50	120.94	7.03	127.93	8.85	75.55	16.67
	0.0012	128.89	10.16	100.91	15.26	99.53	2.11	63.60	6.63
	0.0020	115.23	0.00	69.59	12.38	84.87	14.07	67.11	0.00
	0.0060	80.96	4.75	67.30	1.46	105.96	6.49	61.87	12.04
	0.0120	85.51	5.26	68.92	4.70	91.65	16.09	48.64	12.29
	0.0200	57.79	9.51	66.91	7.74	89.27	17.51	63.91	9.71
72 h	0.0000	100.00	7.08	100.00	3.12	100.00	19.87	100.00	1.90
	0.0004	102.58	7.41	105.76	8.34	70.79	9.03	104.09	6.85
	0.0008	123.63	4.43	113.27	10.07	57.40	16.49	138.32	6.98
	0.0012	159.96	0.00	167.70	0.00	72.83	13.38	101.07	22.41
	0.0020	74.65	0.00	90.88	3.17	73.01	15.02	130.81	25.41
	0.0060	77.42	20.40	132.04	17.81	80.46	5.41	122.58	12.57
	0.0120	81.29	16.83	96.79	9.07	64.67	8.47	92.52	19.87
	0.0200	89.06	14.93	83.97	4.98	113.61	17.04	98.60	14.70

## Data Availability

Not applicable.

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
