# Peer review of "The Role of Functionalization and Size of Gold Nanoparticles in the Response of MCF-7 Breast Cancer Cells to Ionizing Radiation Comparing 2D and 3D In Vitro Models"

_pharmaceutics, 2023, doi:10.3390/pharmaceutics15030862_

Round 1
Reviewer 1 Report
In this study the authors evaluated the different size and functionalization of AuNPs to sensitize 2D and 3D cancer cells to ionizing radiation.
Effectively AuNPs represent an interesting nanomaterial that has been largely employed alone as well as in combination with other drugs for efficient gene therapy, and /or to sensitize cancer cells to chemotherapy and other treatments (as PTT).
this is a large and consistent study, with very interesting potential.They found that PEG chain is essential in the AuNPs efficiency in the process of sensitivity cells to ionizing radiation
I recommend his publication after considering the following minor/major points:
Minor point (text revision):
1) Since AuNPs have been employed largely in cancer application depending on the functionalization and the drug carried, it would be important to remember that different types of functionalization have been described. In the introduction, the authors should add some sentences to discuss more examples of therapeutic applications of AuNPs that are related with this study (for that, I suggest to consider some recent works 10.3390/pharmaceutics13122067, 10.1038/s41467-021-23250-5, 10.3389/fchem.2019.00167)
2) The authors mention breast cancer spheres and perform experiments with this model (figure 2) without specifying before what this concept refers to. Please, discuss what they mean with “sphere” and provide references https://journals.sagepub.com/doi/pdf/10.1177/1010428319869101, https://breast-cancer-research.biomedcentral.com/articles/10.1186/bcr2106
3) In the discussion section it would be interesting to speculate if the nanoparticles used in this study may be employed to deliver bioactive compounds to enhance the therapeutic effect of ionizing radiation.
4) In the experiments of cell viability (figure 3 and figure 4) the graphs are not easy to understand and interpret. For this reason, I recommend including a table with a resume of values for each condition tested.
5) Do these nanoparticles have some effects on migration and invasion of cancer cells? Discuss the potential that AuNPs have in modulating such fundamental processes. https://www.hindawi.com/journals/jnm/2019/1687340/
6) In 3D culture, the viability seems to increase at the lowest concentration of nanoparticles. I think that it is a non-specific effect due to nanoparticles. please explain in the text why they get this result? (170 % of viability respect the ctrl at 72h)
7) Figure 5 is lacking a description of graphs E,F, G, H in the legend.
8) Nanomedicine is a novel topic and it is expected that the cited literature also consists of recent publications ( last 5 years). However, only â…“ of all references have less than 5-years. Please replace them with new ones, when possible, to change this ratio.
9) Discuss that the enhancement of ROS at short time of treatment (fig 5) may lead the cytotoxic effect due to the triggering of oncogenic signaling pathways by ROS (since they may act as signaling molecules to lead cell death https://doi.org/10.1111/cas.15068
11) Uptake of AuNPs: The evaluation of the results, as stated by the authors, was very difficult, since the results are heterogeneous and do not provide any possible understanding of the mechanism. I recommend that authors stated the following concept: “Since the evaluation of uptake has been not conclusive with this experiment, the following step would be functionalized AuNPs with fluorescent probe to better detect their localization, as well as to perform experiment with chemical inhibitors of endocytosis to assess the molecular mechanisms''
12) Mention with a sentence that the effect of ionizing radiation that leads to cell death may be due to the generation of heath, a phenomenon known as hyperthermia”. Indeed, AuNPs ,as well as other metal based NPs may favor this process due to their chemical properties.
13) The authors should stress with more emphasis why there are so many differences between 2D and 3D models after ionizing radiation (fig 8A vs 8B). It seems that 3D models are less responsive. This may appear as a failure of the system proposed in this study.
Major revision:
14) The authors performed the experiments of cytotoxic evaluation only in MCF7 cancer cell lines. It would be important to evaluate the effect of nanoparticles in not-tumor cell lines; indeed, nanomaterials should affect specifically tumor cells but not healthy tissues. If the nanomaterials are not toxic in non-cancer models, it would be encouraging regarding clinical application.
15) Some markers related to cell death (e.i. caspase 3) and oxidative stress (e.i. NRF2 or other) should be analyzed by western blot after treatment with AuNPs + radiation to corroborate the results found by the authors (cell viability and ROS production)
Reviewer 2 Report
The authors here presented the study of AuNP as a radiosensitiser on 2D and 3D MCF7 breast cancer models. AuNPs of both 10 nm and 30 nm were used and the surface was decorated with either PEG800 or PEG2000. The authors first investigated the effect of AuNP on ROS generation in both models. Using the parameters from the experiment, the authors then sought to study the radiosensitivity of AuNP based on a clonogenic assay and the DNA damage repair marker, rH2AX. The manuscript is of interest to the audience that would like to use nanoparticles as radiosensitisers, which is a pressing topic in the field. However, some flaws in the experimental design and ambiguity of the wording compromised the quality of the current manuscript. I suggest the authors consider the following modification of the current manuscript before further submission:
1. The authors should mention which multiple comparison they have used after one-way ANOVA. Also, they should clarify the setting of p-value. The current setting (p < 0.05) will cause type I error inflation and result in flaw interpretation of results. Also, some of the results contain more than one variable (e.g. incubation time vs different AuNPs), the authors should consider using two-way ANOVA to extract interaction information (e.g., how does the viability from different AuNP group change with time?) from these data.
2. Internalisation analysis was conducted by studying the SSC from FACS but did not give any meaningful result. The authors should consider conducting TEM to check the cellular AuNP uptakes.
3. The current design of clonogenic assay makes it hard to interpret the radiosensitisation effect as additive or synergy. The authors should include the AuNP-only group. In addition, the authors should consider including more radiation doses (e.g. 4, 6, 8, and 10 Gy) to calculate the sensitizer enhancement ratio. Also, there is no description of how the clonogenic assay on 3D model was conducted.
4. The rH2AX immunofluorescence staining looks very blurred. I am not sure if this is due to poor conversion from the uploading website or inherent in the staining. A proper rH2AX staining should have clear foci and those foci are quantifiable. The authors should consider quantifying the rH2AX foci per cell to support the FACS result.
5. Please include scale bars on the microscopic images.
6. The discussion and conclusion are very long and contain a lot of non-relevant or repetitive information, which can confuse the readers. Some of the claims are also very overstretched. For example, line 642-643: The differences in the distribution of AuNPs were noticed comparing 2D and 3D models considering the uptake and ROS generation. However, there was no valid study in comparing the distribution of the AuNPs in the manuscript. Also, there was a lack of in-depth discussion about the 3D results and the different ROS formation from AuNPs with different PEG lengths. A summary of the results will be nice to recap the readers about the current finding.
7. Introduction and discussion: the authors should include more previous studies on using AuNP as radiosensitisers. The current citation is very non-specific and fails to relate to the manuscript. For example, the description for references 28 and 29 is very non-specific. Reference 22 showed that a higher concentration of AuNPs resulted in higher ROS production, but this is not consistent with the results from the manuscript.
8. The authors should consider moving some of the results (such as Figure 3, 4, 5) from pilot studies to supplementary.
9. Was the control group stained with rH2AX used to gate the result in Figure 9?
10. The 3D model result of Figure 6 will be benefited from a fixed y-axis scale from 0-6.
11. Figure 3B, the 0 concentration of PEG200-AuNP10 nm showed very low viability. Is there any explanation for that?
12. The authors should conduct more characterisation of their 3D model. From Figure 2, it does not really look like a properly integrate 3D tumour spheroid to me. To demonstrate it is a proper 3D tumour spheroid, the authors can provide the time-course growth curve based on diameters or immunohistology of spheroid cross-section with structural stain (H&E), proliferation or hypoxic markers. This will also help in interpreting some of your 3D data.
13. Previous systematic review of preclinical nanoparticle research showed very low percentage (<5%) of the nanoparticles will end up in tumour when administrated from the general route (through the blood). It will be nice for the authors to discuss based on their results, what would be the best design of nanoparticles for radiotherapy and the best treatment regimen considering the low uptake efficacy.
Round 2
Reviewer 1 Report
The authors improved the discussion of the manuscript, and have taken into account mostly of concerns raised by this reviewer.
Author Response
Dear Editor,
Thank You very much for the review. We appreciate the reviewer's effort, which helped us upgrade the manuscript's text and make it much clearer. All authors assure that no part of the results has been published before by any authors or others.
2 Response to the reviewer:
Thank you for the positive response.
Reviewer 2 Report
I appreciate the reply from the authors but unfortunately, I do not think the authors have answered all my comments properly. In addition, some of their answers to the other reviewer are also not very precise (e.g. reason for 3D more radioresistance). I suggest the authors take time to re-answer the following points:
1. two-way ANOVA: the authors stated that they are not interested in the interaction between two factors but in the experimental design, they have tried more than one factor (different kinds of AuNPs), such as concentration, and incubation time. The authors should try to relate these other factors to the clinical or in vivo radiosensitisation design of AuNPs.
2. It is still unclear to me how the SF is calculated in the 3D model. Does the sphere get dissociated into single cells for plating? Also, the authors stated that they would like to test hyperfractionation regimen for the 2 Gy dose. Any clue that how AuNPs should be administered during the hyperfractionation regimen. For example, will the patient need daily administration of AuNPs?
3. Was the quantification of rH2AX using the microscopic method the same as the 2D culture? Do the authors see any difference in the rH2AX expression on the centre and the peripheral part of the 3D spheres?
4. Citation 8 and 9, the authors should consider adding some conclusions from these references. What do they actually find out and why it is relevant to the current research?
5. I do not think time limitations and preliminary studies are valid reasons for a non-detailed study. After all, studies that are too preliminary should not be considered for publication. I suggest the authors take time to characterise their 3D models and do more research on why 3D models perform differently compared to 2D models in terms of radiobiology.
